# The Influence of the Grapevine Bacterial and Fungal Endophytes on Biomass Accumulation and Stilbene Production by the In Vitro Cultivated Cells of *Vitis amurensis* Rupr.

**DOI:** 10.3390/plants10071276

**Published:** 2021-06-23

**Authors:** Olga A. Aleynova, Andrey R. Suprun, Nikolay N. Nityagovsky, Alexandra S. Dubrovina, Konstantin V. Kiselev

**Affiliations:** Laboratory of Biotechnology, Federal Scientific Center of the East Asia Terrestrial Biodiversity, Far Eastern Branch of the Russian Academy of Sciences, 690022 Vladivostok, Russia; SUPRUN@biosoil.ru (A.R.S.); niknit1996@gmail.com (N.N.N.); dubrovina@biosoil.ru (A.S.D.); kiselev@biosoil.ru (K.V.K.)

**Keywords:** endophyte, bacteria, fungi, cell culture, grape, *Vitis amurensis*, resveratrol, stilbene, stilbene synthase (STS), phenylalanine ammonia lyase (PAL)

## Abstract

Plant endophytes are known to alter the profile of secondary metabolites in plant hosts. In this study, we identified the main bacterial and fungal representatives of the wild grape *Vitis amurensis* Rupr. microbiome and investigated a cocultivation effect of the 14 endophytes and the *V. amurensis* cell suspension on biomass accumulation and stilbene biosynthesis. The cocultivation of the *V. amurensis* cell culture with the bacteria *Agrobacterium* sp., *Bacillus* sp., and *Curtobacterium* sp. for 2 weeks did not significantly affect the accumulation of cell culture fresh biomass. However, it was significantly inhibited by the bacteria *Erwinia* sp., *Pantoea* sp., *Pseudomonas* sp., and *Xanthomonas* sp. and fungi *Alternaria* sp., *Biscogniauxia* sp., *Cladosporium* sp., *Didymella* sp. 2, and *Fusarium* sp. Cocultivation of the grapevine cell suspension with the fungi *Didymella* sp. 1 and *Trichoderma* sp. resulted in cell death. The addition of endophytic bacteria increased the total stilbene content by 2.2–5.3 times, while the addition of endophytic fungi was more effective in inducing stilbene accumulation by 2.6–16.3 times. The highest content of stilbenes in the grapevine cells cocultured with endophytic fungi was 13.63 and 13.76 mg/g of the cell dry weight (DW) after cultivation with *Biscogniauxia* sp. and *Didymella* sp. 2, respectively. The highest content of stilbenes in the grapevine cells cocultured with endophytic bacteria was 4.49 mg/g DW after cultivation with *Xanthomonas* sp. The increase in stilbene production was due to a significant activation of phenylalanine ammonia lyase (*PAL*) and stilbene synthase (*STS*) gene expression. We also analyzed the sensitivity of the selected endophytes to eight antibiotics, fluconazole, and *trans*-resveratrol. The endophytic bacteria were sensitive to gentamicin and kanamycin, while all selected fungal strains were resistant to fluconazole with the exception of *Cladosporium* sp. All endophytes were tolerant of *trans*-resveratrol. This study showed that grape endophytes stimulate the production of stilbenes in grape cell suspension, which could further contribute to the generation of a new stimulator of stilbene biosynthesis in grapevine or grape cell cultures.

## 1. Introduction

Endophytes are microorganisms that inhabit the tissues of living plants without any negative consequences for plant growth and development. Plant endophytes are usually presented by fungi and bacteria and less often algae and viruses [1]. In recent years, plant endophytes have attracted more attention from researchers in terms of their diversity and application for improving plant properties or plant disease protection. Endophytes inhabiting grapes are no exception. Grapes are one of the most important fruit crops in the world but are largely affected by a large number of pathogens that cause diseases before and after harvest, affecting production, processing, exports, and the quality of the fruit.

There are many studies devoted to the microbiome of grapes of the European grape group [2,3,4]. It has been shown that some grape endophytes are able to prevent infection, inhibit grape pathogens, increase abiotic stress resistance, and increase the content of nutrients in berries [5,6,7,8]. Additionally, it has been found that exposure of grape cells to endophytic fungi (*Alternaria*
*alternate* and *Epicoccum nigrum*) and a dual culture system differentially affected total anthocyanin concentrations and phenylalanine ammonia lyase activities [9]. It has also been shown that fungal endophytes form new metabolite patterns in grape cells during double culture and showed fungal strain specificity [10].

While there is a number of studies that examined grapevines of European grapes (*Vitis vinifera*), there is not enough information about the endophytes of the Asian group of grapes, e.g., *Vitis amurensis* Rupr. This species is of interest because it grows in the wild, and it is a leader in terms of the content of valuable biologically active substances (stilbenes) among grape species. Furthermore, *V. amurensis* exhibits a high resistance to low temperatures and microbial diseases [11,12]. Knowledge on the endophytic microorganisms inhabiting the highly resistant grapevine *V. amurensis* could help develop new approaches for prevention of the emergence and spread of grape diseases.

*V. amurensis* is highly resistant to such widespread grapevine diseases as powdery mildew, the pathogen *Oidium tuckeri* (a teleomorph of *Uncinulanecator*) [13], grape white rot, and anthracnose [14]. The biologically active compounds found in the grape stem of *V. amurensis* are capable of suppressing pathogenic bacteria, such as *Streptococcus mutans* and *Streptococcus sanguis* [15]. Chinese scientists have described a new pathogenic fungus, *Fusarium avenaceum*, which affects the fruits of *V. amurensis* [16]. Recently, it has been shown that the endophytic fungus *Albifimbria verrucaria* isolated from Amur grapes was active against *Botrytis cinerea*, causing gray mold disease in grapes [17]. To the best of our knowledge, there are no other studies analyzing the microorganisms inhabiting and affecting *V. amurensis*.

In the present work, we focused on the influence of basic endophytic bacteria and fungi of wild grapes (*V. amurensis*) on the biosynthesis of stilbenes in *V. amurensis* cell suspension culture. Stilbenes are a relatively small group of naturally occurring phenolic compounds found in a number of unrelated plant families, such as peanuts (Fabaceae), pine (Pinaceae), or grapes (Vitaceae). Grapevine is a leader in the content of resveratrol among other plants. The most famous stilbene is *trans*-resveratrol or *t*-resveratrol (3,5,4′-trihydroxy-*trans*-stilbene), which is the main precursor in the biosynthesis of other stilbenes [18] and possesses a wide spectrum of biological activities [19]. It has been established that *t*-resveratrol is able to prevent the occurrence and development of cardiovascular and oncological diseases, exhibits antiallergic effects, and slows down the aging process [19,20]. Additionally, stilbenes are important for plant protection against microbial pathogens [21]. In this research, we studied the effect of the main endophytic bacteria (*Agrobacterium* sp., *Bacillus* sp., *Curtobacterium* sp., *Erwinia* sp., *Pantoea* sp., *Pseudomonas* sp., *Xanthomonas* sp.) and endophytic fungi (*Alternaria* sp., *Didymella* sp. 1, *Didymella* sp. 2, *Cladosporium* sp., *Fusarium* sp., *Trichoderma* sp., and *Biscogniauxia* sp.) on the cell growth, production, and biosynthesis of stilbenes in *V. amurensis* cell culture.

## 2. Results

### 2.1. Identification of V. amurensis Endophytes

We selected and identified the main representatives of the microbiome of *V. amurensis*. Analysis of the nucleotide sequences of *16S rRNA* genes (for bacteria) and sequences of internal transcribed spacers *ITS1* (for fungi) revealed that the isolated strains are representatives of the endophytic bacteria genera *Agrobacterium* sp., *Bacillus* sp., *Curtobacterium* sp., *Erwinia* sp., *Pantoea* sp., *Pseudomonas* sp., and *Xanthomonas* sp., as well as endophytic fungi genera *Alternaria* sp., *Didymella* sp. *1*, *Didymella* sp. *2*, *Cladosporium* sp., *Fusarium* sp., *Trichoderma* sp., and *Biscogniauxia* sp. (Table 1).

### 2.2. The Influence of V. amurensis Endophytes on the Fresh Biomass Accumulation in V. amurensis Cell Culture

We added the selected endophytic bacteria and fungi to a cell suspension culture of *V. amurensis* (V7) and estimated the accumulation of fresh biomass and stilbene content. The addition of *Agrobacterium* sp., *Bacillus* sp., and *Curtobacterium* sp. did not significantly affect the accumulation of fresh biomass of the grapevine cell suspension culture after 2 weeks of cocultivation (Figure 1). However, the addition of bacteria *Erwinia* sp., *Pantoea* sp., *Pseudomonas* sp., and *Xanthomonas* sp. and fungi *Alternaria* sp., *Biscogniauxia* sp., *Cladosporium* sp., *Didymella* sp. *2*, and *Fusarium* sp. significantly inhibited the accumulation of the cell culture fresh biomass by 1.7–2.7 times (Figure 1). Cocultivation of the V7 cells with the endophytic fungi *Didymella* sp. *1* and *Trichoderma* sp. resulted in cell death (Figure 1).

### 2.3. The Effect of the Endophytic Bacteria and Fungi on Stilbene Content in the V. amurensis Cell Suspension

High-performance liquid chromatography (HPLC) analysis revealed that the total stilbene content in the V7 cell suspension increased by 2.2–16 times after the addition of endophytes (Figure 2). The highest stilbene content was 2.2–4.5 mg/g DW, which occurred under the influence of bacteria *Curtobacterium* sp., *Erwinia* sp., *Pantoae* sp., *Pseudomonas* sp., and *Xanthomonas* sp. The highest stilbene content in the V7 cell culture after bacterial treatment was 4.5 mg/g DW, which occurred during cocultivation with bacteria *Xanthomonas* sp. (Figure 2). However, a more significant increase in stilbene content to 6.95–13.76 mg/g DW was detected after culture cocultivation with the endophytic fungi *Biscogniauxia* sp., *Cladosporium* sp., *Didymella* sp. 1, and *Didymella* sp. 2 (Figure 2).

Then, we analyzed the composition and content of individual stilbenes in the culture samples after cocultivation with the endophytes (Table 2). A distinctive feature of the treatment with bacteria and fungi was that the relative content of glycosylated stilbenes decreased significantly in the samples. In the control cell culture, the total content of di-glucoside *trans*-resveratrol, *trans*-piceid, and *cis*-piceid was about 64% of the total amount of stilbenes, while it was 28–57% when treated with bacteria and only 13–31% when treated with fungi (Table 2).

The second feature was a drastic increase in the content of *trans*-resveratrol after the addition of bacteria and fungi by 4.8–16.4 and 4.9–98.5 times, respectively. The highest *trans*-resveratrol content was 2.9 mg/g DW, which was detected in grapevine cells cultivated with an endophytic fungus, *Biscogniauxia* sp. (Table 2). Among the bacteria, the most stimulating effect on the content of *trans*-resveratrol was exerted by the cultivation with bacteria *Pantoae* sp. leading to a content of 0.5 mg/g DW (Table 2). The content of viniferins (*epsilon*-viniferin and *delta*-viniferin) was also substantially increased after the addition of bacteria and fungi by 2.8–8.4 and 4.8–32.1 times, respectively. For example, when cultured with the endophytic fungus, *Biscogniauxia* sp., the amount of viniferins reached 9 mg/g DW, which was 65% of all detected stilbenes. At the same time, the amount of viniferins in the control V7 cell culture without bacterium or fungi treatment was only 33% (Table 2).

### 2.4. VaPAL and VaSTS Gene Expression in V. amurensis Cells with the Addition of Endophytes

Then, we analyzed *VaPAL* and *VaSTS* gene expression after cocultivation with the endophytic bacteria and fungi. We also analyzed the effect of *Agrobacterium sp.* on the expression of the *VaPAL* and *VaSTS* genes to use it as a negative control, as this bacterium did not significantly increase stilbene content in the grapevine cells. We found that the addition of all selected endophytic fungi led to a significant activation of the *VaPAL1-VaPAL4* genes by 2.6–44 times compared with untreated cells V7 (Figure 3a). The addition of the endophytic bacteria differently activated the expression of each *VaPAL* gene. The expression of the *VaPAL1*-*VaPAL3* genes significantly increased by 1.7–17 times after the addition of *Erwinia* sp. and *Pantoae* sp. bacteria. The addition of *Pantoae* sp. significantly increased the expression of *VaPAL4* (Figure 3a). The expression of the *VaPAL3* gene was significantly increased when the endophytic bacteria *Agrobacterium* sp. and *Pseudomonas* sp. were added to the V7 cell culture. The addition of bacteria *Xanthomonas sp.* significantly increased only the expression of the *VaPAL2* gene (Figure 3a).

The expression of *VaSTS1*-*VaSTS5* genes significantly increased after cocultivation with the endophytic bacteria *Erwinia* sp. and *Pantoae* sp. and all selected endophytic fungi (Figure 3b,c). The expression of the *VaSTS2* gene significantly increased with the addition of all selected bacteria (Figure 3b). With the addition of all selected endophytes, *STS7* gene expression increased most significantly in comparison with other *STS* genes. The greatest expression of this gene was 32 times higher than in the untreated V7 cells and was observed after the addition of the fungus *Biscogniauxia* sp. (Figure 3c). The increase in *STS8* gene expression was observed with the addition of *Pantoae* sp. bacteria and fungi *Cladosporium* sp. and *Didymella* sp. 2. The addition of *Biscogniauxia* sp. and *Cladosporium* sp. fungi significantly increased *STS9* and *STS10* gene expression. We also observed an increase in the *VaSTS10* gene expression after the addition of bacteria *Pantoae* sp. (Figure 3c).

### 2.5. Sensitivity of the Grape Endophytes to Antibiotics, Fluconazole, and Resveratrol

We tested the sensitivity of the bacteria to antibiotics and to *trans-*resveratrol, which is stilbene with known antibacterial properties [22,23]. We placed antibiotic-soaked paper disks on plates with bacteria and evaluated the bacterial growth line around the paper disk after 2 days. We observed that gentamicin and kanamycin inhibited the growth of all selected endophytic bacteria (Table 3). Rifampicin also inhibited the growth of all endophytic bacteria, except for *Pantoae* sp. The growth of *Agrobacterium* sp. was also suppressed by antibiotics, such as ampicillin, cefotaxime, spectinomycin, and tetracycline. The growth of *Erwinia* sp. was suppressed by the antibiotic cefotaxime, and the growth of *Pseudomonas* sp. was suppressed by spectinomycin and tetracycline (Table 3). All endophytic bacteria were resistant to *trans-*resveratrol (Table 3).

The sensitivity of fungi to fluconazole and *trans*-resveratrol was evaluated in a similar way. Only one fungus, *Cladosporium* sp., was found to be sensitive to fluconazole (Table 3). The rest of the endophytic fungi changed color when growing on a paper disk soaked in fluconazole, but no visible growth-restriction zone was observed (Table 3). Additionally, all endophytic fungi were resistant to *trans-*resveratrol (Table 3). The fact that none of the endophytes were inhibited by *trans-*resveratrol suggests that these are indeed typical endophytes of *V. amurensis* and developed protective mechanisms against this stilbene.

## 3. Discussion

The content of stilbenes in the plant material did not exceed 0.01% of the dry mass of cells, which could increase the cost of producing stilbenes on an industrial scale. Today, there are many ways in biotechnology to stimulate the biosynthesis of stilbenes, but these approaches are mainly based on using chemicals that are often unfavorable to human health. In recent years, investigation and application of the plant endophytes and endophyte-based preparations have been actively developed to improve crop properties, i.e., abiotic and biotic stress resistance, yield and fruit quality, and the content of valuable secondary metabolites.

To date, several works have been devoted to studying the effect of grape endophytic fungi on the metabolism and performance in grapevine cell cultures. It has been shown that the cocultivation of grape cells with different strains of fungi led to the appearance of new metabolites (from 1 to 11) specific for the strain/genus of endophytic fungi, previously not characterized in grape cell culture [24]. Furthermore, the addition of fungal endophytes to grape cell culture resulted in changed primary and secondary metabolism. The total sugar content, titrated acidity, total soluble protein content, total flavonoids and phenols, and malondialdehyde, as well as the activity of antioxidant enzymes, guaiacol-dependent peroxidase, superoxide dismutase, and *PAL*, were changed [10]. Fungal endophytes can produce different elicitors or other signaling molecules, which in turn causes different metabolic changes, so the oxidative response may be a result of a common cell response to the changes in metabolism mediated by endophytes in the host plants [25].

Recently, Chinese scientists found that the addition of endophytic fungi to grape cell culture had different effects on the total concentration of anthocyanins and the activity of *PAL* in grape cells. When the strains of the fungi *Alternaria alternata* and *Epicoccum nigrum* were cocultured with grape cells, the anthocyanin content increased by 74% and 28%, respectively, while the addition of another strain of *A. alternata* reduced the anthocyanin content by 19% [9]. Thus, during the cocultivation of grape cells together with fungal endophytes, protective reactions occur in the grape cells leading to metabolic changes in the grape cells.

This study firstly examined the influence of the main representatives of the endophytic bacteria and fungi of wild grape *V. amurensis* on cell growth and the production of resveratrol and its derivatives. The highest total content of stilbenes, particularly *trans*-resveratrol, was observed in the *V. amurensis* cell culture with the addition of the fungi *Biscogniauxia* sp., *Cladosporium* sp., and *Didymella* sp. 2, which correlated with a significant increase in the expression of most analyzed *PAL* and *STS* genes. The stilbene level reached 13.8 mg/g DW (or 1.35% DW) after cocultivation with endophytic fungi and 4.5 mg/g DW (or 0.45% DW) after cocultivation with endophytic bacteria. This significant increase was due to a strong increase in the biosynthesis of *trans-*resveratrol and its oligomers, viniferins.

It is important to note that this resulting stilbene level was one of the highest observed thus far for plant cell cultures [26]. For example, at 18 h after UV-C treatment, the stilbene content in *Arachis hypogaea* callus cultures reached 0.017 mg/g FW or approximately 0.3 mg/g DW [27]. The culture that had been inoculated with the plant pathogen *Botryodiplodia theobromae* stilbene content reached 0.023 mg/g FW or approximately 0.5 mg/g DW [28]. In *V. vinifera* cell suspension culture, the stilbene level was 2.1 mg/g DW at 2 days after UV-C and 100 µM MeJa treatment [29].

The reached stilbene content in the present investigation was higher than the stilbene content in the leaves of normally cultivated grapevine and after UV treatment (0.04–0.95 mg/g DW) [30,31]. However, the stilbene content in our experiments was lower than the stilbene content reached after the application of cyclic oligosaccharides (cyclodextrins (CDs)), separately or in combination with methyl jasmonate or some other plant stress hormones [32,33]. After using these inducing agents, only *t*-resveratrol content in the *V. vinifera* cell cultures reached 35–155 mg/g DW [32,33].

It has also been shown that almost all endophytic bacteria were sensitive to antibiotics, such as gentamicin and kanamycin. Thus, it is possible to use these bacteria for the induction of stilbene production in *V. amurensis* cell culture. Moreover, the fungus *Cladosporium* sp. could be a promising supplement for the induction of stilbene biosynthesis, since it significantly activated stilbene production and was sensitive to fluconazole.

## 4. Conclusions

For the first time, this study examined the influence of the main representatives of the endophytic bacteria and fungi of wild grape *V. amurensis* on cell growth and production of resveratrol and its derivatives. The addition of endophytic bacteria increased the total stilbene content by 2.2–5.3 times, while the addition of endophytic fungi was more effective in inducing stilbene accumulation by 2.6–16.3 times. The highest content of stilbenes in the grapevine cells cocultured with the endophytic fungi was 13.63 and 13.76 mg/g of the cell dry weight (DW) after cocultivation with *Biscogniauxia* sp. and *Didymella* sp. 2, respectively. The highest content of stilbenes in the grapevine cells cocultured with the endophytic bacteria was 4.49 mg/g DW after cultivation with *Xanthomonas* sp. The increase in stilbene production was due to a significant activation of *VaPAL* and *VaSTS* gene expression.

Thus, cocultivation of grape cell culture with the natural endophytes of *V. amurensis* could be considered as a new natural approach for the activation of stilbene biosynthesis, which could be used for industrial production of stilbenes in grape cell cultures. The development and active application of such approaches would undoubtedly contribute to the transition to a highly productive and environmentally friendly agriculture.

## 5. Materials and Methods

### 5.1. Plant Material

The stems and leaves of wild-growing *V. amurensis* (excised young stems 7–8 cm long with three healthy leaves from 2 adult plants) were sampled from a nonprotected natural population near Vladivostok, Russia (the southern Primorsky region of the Russian Far East, longitude 43.2242327 and latitude 131.99112300). The leaves and stems were collected from two grape plants located at a distance of 1 km from each other. The material was collected over 3 years in June (young leaves and stems), July (mature leaves and stems), and September (leaves that have become drier and changed color) 2018–2020. Each plant sample was delivered within half an hour to a temperature of 18–23 °C to the laboratory in sterile flasks and used for endophyte isolation.

### 5.2. Isolation and Identification of the Endophytic Bacteria and Fungi

A 1.5 g amount of the leaf and stem tissue was washed with soap. Then, under sterile conditions, they were soaked in 75% ethanol for 2 min and then in 10% hydrogen peroxide solution for 1 min and washed 5 times with sterile water. To check the efficacy of this method of surface sterilization, 100 µL of the last wash water was incubated on potato dextrose agar (PDA, Neogene, UK) for fungi and on the R2A medium for bacteria [34]. The plates were inspected for the absence of colony growth. In a sterile mortar, the leaf and stem tissue of *V. amurensis* was ground to a homogeneous state, and the juice was squeezed out. Then, 100 µL of the juice was applied to Petri dishes with PDA and R2A medium. After 3 days, the colonies cultivated were picked out and transferred carefully to a new sterile plate for reculturing.

DNA of some of the distinct bacteria and fungi colonies was isolated by the hexadecyltrimethylammonium bromide (CTAB) method with modifications [35]. Bacterial *16S rRNA* gene sequences were amplified by universal bacterial primers for the amplification of approximately 1500 bp *16S* PCR products (8F, 5′ AGA GTT TGA TCM TGG CTC AG and 1522R, 5′ AAG GAG GTG ATC CAR CCG CA) [36]. The universal primers 5′ AGG AGA AGT CGT AAC AAG G and 5′ TCC TCC GCT TAT TGA TAT GC were used for the amplification of approximately 580 bp *ITS1* PCR products [37]. PCR products were sequenced using an ABI 3130 Genetic Analyzer (Applied Biosystems, Foster City, CA, USA) according to the manufacturer’s instructions. The Basic Local Alignment Search Tool (BLAST) program was used for sequence analysis. Multiple sequence alignments were performed using the Clustal X program [38]. A sequence identity of ≥99% is considered as a sufficient threshold value for taxonomic identification.

### 5.3. Treatment of Grape Cells with Endophytic Bacteria and Fungi

The V7 callus culture was established in 2017 from young stems of the mature *V. amurensis* plants as described [39]. A 2.0 g amount of V7 cell culture (each callus was weighed using an electronic balance) was cultured in 50 mL of liquid Murashige and Skoog-modified W_B/A_ medium supplemented with 0.5 mg/L 6-benzylaminopurine (B) and 2 mg/L α-naphthaleneacetic acid constantly stirring in the orbital shaker in the dark for 7 days [40]. Then, 100 μL of endophytic bacterial suspension and 100 mg of endophytic fungus were added to the flask. Endophytes were preliminarily grown in 50 mL of liquid media R2A for bacteria and potato dextrose media for fungi in the orbital shaker for 3 days (130 rpm, 23 °C). Endophytes were cultivated with V7 grape cell culture for stilbene analysis for 3 days and for biomass evaluation for 7 days. Recultivation of the endophytic strains of bacteria and fungi was carried out once or twice before cocultivation with the grape cells. The data for stilbene and fresh biomass accumulation in the cell cultures were obtained from three independent experiments with three replicates each.

### 5.4. Total RNA Extraction, Reverse Transcription, and qRT-PCR

RNA isolation was carried out from the week-old cell-cultured V7 with 3-day cocultivation with endophytes. Total RNA isolation was performed using the CTAB-based-based protocol [41]. Complementary DNAs were synthesized as described [42]. The reverse transcription products were then amplified by PCR and verified in the absence of DNA contamination using primers listed in Appendix A. The qRT-PCRs were performed with EvaGreen Real-Time PCR (Biotium, Hayward, Berkeley Heights, NJ, USA) as described [43] using cDNAs V7 callus culture and two internal controls (*GAPDH* and *Actin*), which were selected in previous studies as relevant reference genes for real-time PCRs for grapevine [44]. The expression was calculated by the 2^−ΔΔCT^ method [45]. All GenBank accession numbers and primers are listed in Appendix A.

### 5.5. High-Performance Liquid Chromatography

The dried and powdered V7 culture samples (100 mg) were extracted with 95% EtOH (2 mL) for 2 h at 60 °C. After extraction, we purified them with OlimPeak syringe filters, nylon, with a pore size of 0.45 µm and a diameter of 13 mm (Teknokroma, Barcelona, Sant Cugat del Vallés, Spain), and then used them for HPLC analysis. The measurement for each sample was repeated 3 times.

Identification and quantification of all stilbenes was performed using an HPLC LC-20AD XR analytical system (Shimadzu, Japan) and commercially available standards. DAD data were recorded in the 200–500 nm range, and chromatograms for quantification were acquired at 310 nm. The chromatographic separation was performed on a Shim-pack GIST C18 column (150 mm, 2.1 nm i.d., 3 µm part size; Shimadzu, Japan). Extracts from cells cultures were separated using 0.1% formic acid and acetonitrile as mobile phases A and B, respectively, with the following elution profile: 0 to 35 min 0% of B; 35 to 40 min 40% of B; 40 to 50 min 50% of B; 50 to 65 min 100% of B. A 3 μL volume of the sample extract was injected with a constant column temperature maintained at 40 °C and a flow rate maintained at 0.2 mL/min. The contents of stilbenes were determined by using the external standard method using the five-point regression calibration curves built with the available standards. The analytical standards *trans*-resveratrol, *trans*-piceid, and *trans*-piceatannol were obtained from Sigma-Aldrich (St. Louis, MO, USA), and *d*-viniferin was obtained from Panreac AppliChem (GmbH, Darmstadt, Germany). *Cis* isomers of resveratrol and piceid were obtained under sunlight exposure of the respective standard solution containing the *trans*-isomer as reported earlier [31].

### 5.6. Antibiotic Susceptibility Analysis

To analyze sensitivity to antibiotics, 100 μL of a suspension of endophytic bacteria and 1 cm^2^ of endophytic fungi were placed on Petri dishes with agar culture media R2A for bacteria and PDA for fungi [22]. Paper disks 1 cm in diameter soaked in antibiotic solution were placed on top. Antibiotic concentrations were ampicillin—50 mg/L, chloramphenicol—35 mg/L, cefotaxime—250 mg/L, gentamicin—50 mg/L, kanamycin—50 mg/L, rifampicin—50 mg/L, spectinomycin—150 mg/L, and tetracycline—40 mg/L. A similar method was used to analyze the sensitivity of endophytes to resveratrol at concentrations of 1 mM.

### 5.7. Statistical Analysis

The data are presented as mean ± standard error (SE) and were tested by Student’s *t-*test. The 0.05 level was selected as the point of minimal statistical significance in all analyses. Three independent experiments were performed for each type of experiment.

## Figures and Tables

**Figure 1 plants-10-01276-f001:**
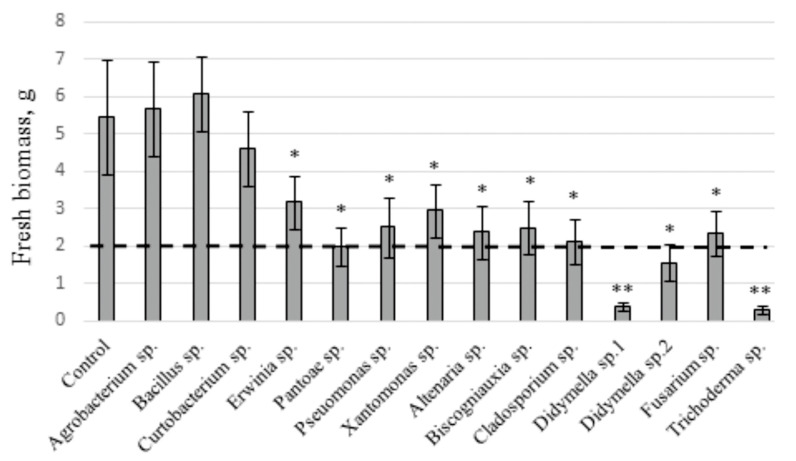
Accumulation of fresh biomass in the V7 cell suspension culture of *Vitis amurensis* after one week of cocultivation with the *V. amurensis* endophytic bacteria or fungi. The dotted line depicts an inoculum mass of 2 g. * *p* < 0.05; ** *p* < 0.01 versus values of fresh biomass accumulated in the V7 cell under the control conditions without bacteria and fungi.

**Figure 2 plants-10-01276-f002:**
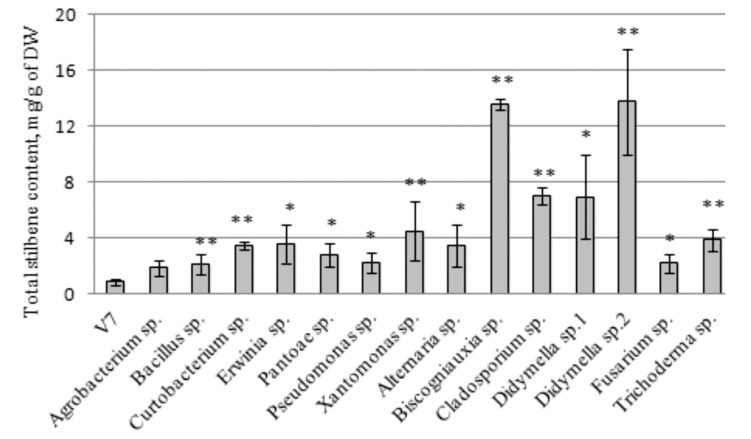
The total stilbene content (mg/g of the cell dry weight (DW)) in the V7 cell suspension culture of *Vitis amurensis* V7 after 3 days cocultivation with endophytic bacteria or fungi. * *p* < 0.05; ** *p* < 0.01 versus values of stilbene accumulation in the V7 cell cultivated under control conditions without bacteria and fungi.

**Figure 3 plants-10-01276-f003:**
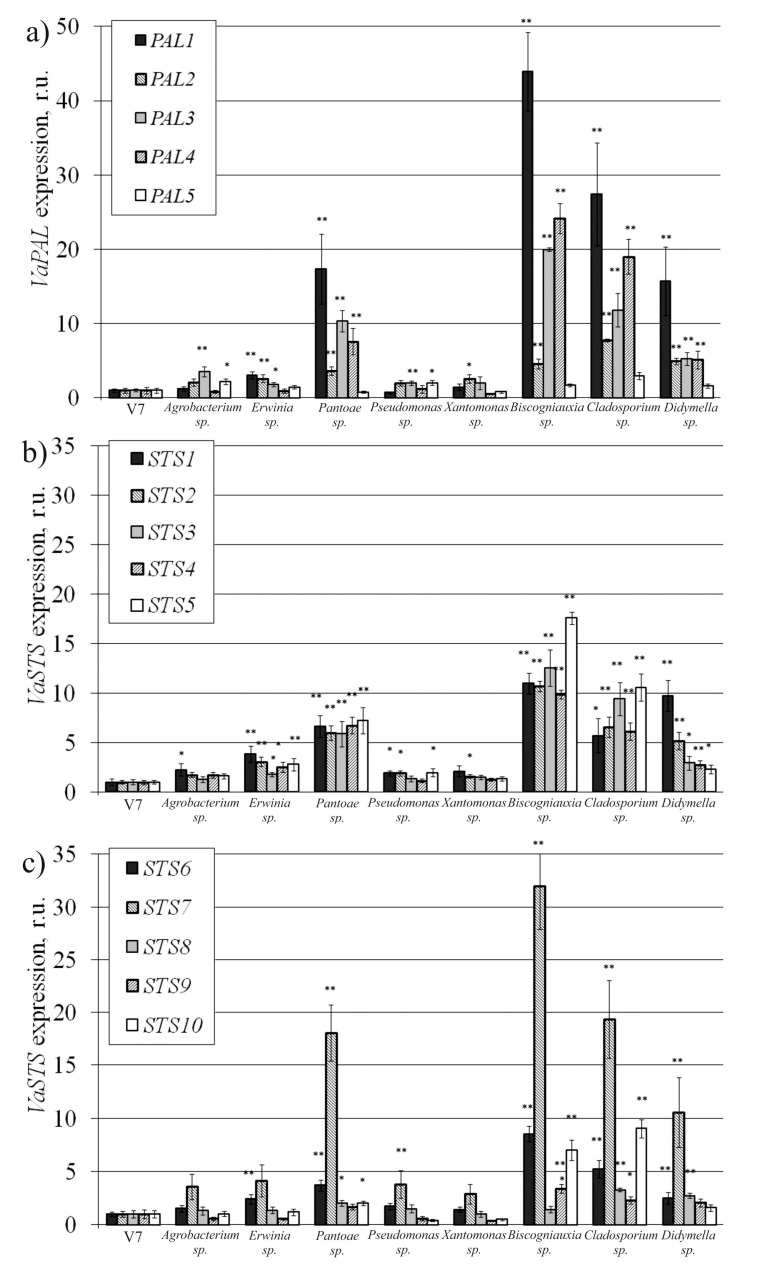
*VaPAL* (**a**) and *VaSTS* (**b**,**c**) gene expression in the V7 cell suspension culture of *Vitis amurensis* after 3 days of cocultivation with the endophytic bacteria *Agrobacterium sp.*, *Erwinia sp.*, *Pantoae sp.*, *Pseudomonas sp.*, *Xantomonas sp.* or fungi *Biscogniauxia sp.*, *Cladosporium sp.*, and *Didymella sp.* 2. * *p* < 0.05; ** *p* < 0.01 versus values of fresh biomass accumulation in the V7 cells cultivated under control conditions without bacteria and fungi.

**Table 1 plants-10-01276-t001:** Characteristics of the bacteria and fungi strains isolated from *Vitis amurensis* grape microbiome based on *16S rRNA* (bacteria) gene or internal transcribed spacers *ITS1* (fungi) sequences. The resulting nucleotide sequences were collected using the Staden Package program. The percentage identities of the collected nucleotide sequences were determined using NCBI BLAST (http://blast.ncbi.nlm.nih.gov; accessed date 11 May 2021) and the Nucleotide Blast (nucleotide—nucleotide BLAST) algorithm.

№	Used Gene	Genus and Sequence ID	The Close Species and Sequence ID	Percent Identity
1	*16S rRNA*	*Agrobacterium* (MZ424738)	*Agrobacterium rubi* (MN752429.1)	99.17%
2	*16S rRNA*	*Bacillus* (MZ424739)	*Bacillus thuringiensis* (KU179338.1)	100%
3	*16S rRNA*	*Curtobacterium* (MZ424740)	*Curtobacterium flaccumfaciens* (AJ310414.1)	100%
4	*16S rRNA*	*Erwinia* (MZ424741)	*Erwinia billingiae* (KM408608.1)	100%
5	*16S rRNA*	*Pantoae* (MZ424742)	*Pantoea agglomerans* (MT605813.1)	99.75%
6	*16S rRNA*	*Pseudomonas* (MZ424743)	*Pseudomonas alkylphenolica* (MN813762.1)	99.89%
7	*16S rRNA*	*Xanthomonas* (MZ424744)	*Xanthomonas campestris* (MN108237.1)	99.13%
8	*ITS1*	*Alternaria* (MZ427922)	*Alternaria tenuissima* (KF308883.1)	100%
9	*ITS1*	*Biscogniauxia* (MZ427923)	*Biscogniauxia maritima* (MN341558.1)	100%
10	*ITS1*	*Cladosporium (*MZ427924)	*Cladosporium perangustum* (MT645918.1)	100%
11	*ITS1*	*Didymella* (MZ427925)	*Didymella negriana* (MK100201.1)	100%
12	*ITS1*	*Didymella* (MZ427926)	*Didymella pinodella* (KX869956.1)	100%
13	*ITS1*	*Fusarium* (MZ427927)	*Fusarium tricinctum* (MT446111.1)	100%
14	*ITS1*	*Trichoderma* (MZ427928)	*Trichoderma harzianum* (MT422092.1)	98.97%

**Table 2 plants-10-01276-t002:** The content of stilbenes in the *Vitis amurensis* cell culture V7 after 3 days of cocultivation with endophytic bacteria or fungi, mg/g DW. * *p* < 0.05; ** *p* < 0.01 versus values of stilbene accumulation in the V7 cell cultivated under control conditions without bacteria and fungi.

	Di-glucoside *trans*-Resveratrol	*trans*-Piceid	*trans*-Resveratrol	*epsilon*-Viniferin	*delta*-Viniferin	*cis*-Resveratrol	*cis*-Piceid	*trans*-Piceatannol
*V7, Control*	0.432 ± 0.135	0.104 ± 0.022	0.029 ± 0.008	0.024 ± 0.007	0.255 ± 0.077	0.0004 ± 0.0001	0.0103 ± 0.0061	0
*Agrobacterium* sp.	0.583 ± 0.167	0.136 ± 0.055	0.140 ** ± 0.036	0.279 * ± 0.129	0.802 * ± 0.240	0.0010 ± 0.0004	0.0017 ± 0.0015	0
*Bacillus* sp.	0.823 ± 0.322	0.291 ** ± 0.053	0.141 ** ± 0.049	0.190 ** ± 0.068	0.575 ± 0.213	0.0011 *± 0.0004	0.0693 ± 0.0561	0.0014 ± 0.0014
*Curtobacterium* sp.	3.447 * ± 0.247	0.348 ** ± 0.082	0.181 ** ± 0.043	0.131 ** ± 0.024	1.412 ** ± 0.415	0.0015 ** ± 0.0003	0.1025 ** ± 0.0674	0
*Erwinia* sp.	0.860 ± 0.350	0.243 ± 0.078	0.364 * ± 0.201	1.154 ** ± 0.535	0.940 * ± 0.333	0.0018 * ± 0.0008	0.0143 ± 0.0059	0.0249 * ± 0.0228
*Pantoae* sp.	0.602 ± 0.122	0.194 ± 0.074	0.477 ** ± 0.291	0.744 ** ± 0.204	0.770 ± 0.292	0.0010 ± 0.0003	0.0117 ± 0.0050	0.0013 ± 0.0013
*Pseudomonas* sp.	0.486 ± 0.232	0.230 ± 0.094	0.167 ** ± 0.047	0.440 ** ± 0.158	0.861 * ± 0.304	0.0016 ± 0.0007	0.0084 ± 0.0045	0.0245 ± 0.0245
*Xantomonas* sp.	4.491 * ±2.124	0.364 ** ± 0.148	0.433 ** ± 0.012	0.372 ** ± 0.220	1.958 * ± 1.605	0.0022 ** ± 0.0001	0.0268 ± 0.0268	0
*Alternaria* sp.	0.620 ± 0.255	0.239 ± 0.098	0.338 ** ± 0.117	1.292 ** ± 0.586	0.941 ± 0.469	0.0029 ± 0.0013	0.0501 ± 0.0326	0.0046 ± 0.0046
*Biscogniauxia* sp.	1.265 * ± 0.206	0.477 ** ± 0.106	2.861 ** ± 0.415	3.952 ** ± 0.664	4.981 ** ± 0.335	0.0035 * ± 0.0002	0.0910 ± 0.0162	0
*Cladosporium* sp.	1.258 ** ± 0.178	0.346 ** ± 0.082	0.882 ** ± 0.136	2.964 ** ± 0.132	1.504 * ± 0.866	0.0040 ** ± 0.0002	0.0216 ± 0.0125	0
*Didymella* sp. 1	1.035 ± 0.667	0.427 * ± 0.339	1.180 * ± 1.130	2.082 ** ± 1.872	2.222 * ± 2.060	0.002 ± 0.002	0	0
*Didymella* sp. 2	2.184 ** ± 0.945	0.619 ** ± 0.077	2.796 ** ± 1.856	3.860 ** ± 0.076	3.994 ** ± 0.775	0.005 ** ± 0.001	0.298 ** ± 0.169	0
*Fusarium* sp.	0.487 ± 0.236	0.147 ± 0.078	0.171 * ± 0.073	0.614 ** ± 0.163	0.700 * ± 0.229	0.0173 ** ± 0.0165	0.0295 ± 0.0184	0
*Trichoderma* sp.	0.682 ± 0.024	0.099 ± 0.001	0.127 ** ± 0.015	0.195 ** ± 0.079	2.706 ** ± 0.884	0.0025 ** ± 0.0004	0.0465 ± 0.0268	0.0094 * ± 0.0054

**Table 3 plants-10-01276-t003:** Analysis of the sensitivity of endophytic bacteria and fungi of *Vitis amurensis* to antibiotics and *trans*-resveratrol. The values are given as the distance (cm) from the antibiotic-soaked paper disk to the visible bacterial growth zone. Ap—ampicillin (50 mg/L), Cam—chloramphenicol (35 mg/L), Cf—cefotaxime (250 mg/L), Gent—gentamicin (50 mg/L), Km—kanamycin (50 mg/L), Rf—rifampicin (50 mg/L), Sp—spectinomycin (150 mg/L), Tet—tetracycline (40 mg/L), Fluc-fluconazole (50 mg /L), Res—*trans-*resveratrol (1 mM); n.m.—not measured; * external visible changes of the fungal coloring.

	Antibiotics								Res.	Fluc.
Genus	Ap	Cam	Cf	Gent	Km	Rf	Sp	Tet		
*Agrobacterium*	0.3	0	1	1	1.2	0.5	1	1.6	0	n.m.
*Bacillus*	0	0	0	0.6	0.5	0.6	0	0	0	n.m.
*Erwinia*	0	0	1	0.5	0.6	0.3	0	0	0.2	n.m.
*Pantoae*	0	0	0	0.6	0.8	0	0	0	0	n.m.
*Curtobacterium*	0	0	0	0.8	1	2	0	0.2	0	n.m.
*Xantomonas*	0	0	0	0.6	0.8	0.8	0	0	0	n.m.
*Pseudomonas*	0	0	0	0.6	0.5	0.7	0.6	0.7	0	n.m.
*Fusarium*	n.m.	n.m.	n.m.	n.m.	n.m.	n.m.	n.m.	n.m.	0 *	0.1 *
*Alternaria*	n.m.	n.m.	n.m.	n.m.	n.m.	n.m.	n.m.	n.m.	0 *	0.1 *
*Didymella-1*	n.m.	n.m.	n.m.	n.m.	n.m.	n.m.	n.m.	n.m.	0 *	0 *
*Didymella-2*	n.m.	n.m.	n.m.	n.m.	n.m.	n.m.	n.m.	n.m.	0 *	0 *
*Trichoderma*	n.m.	n.m.	n.m.	n.m.	n.m.	n.m.	n.m.	n.m.	0	0 *
*Cladosporium*	n.m.	n.m.	n.m.	n.m.	n.m.	n.m.	n.m.	n.m.	0 *	0.5
*Biscogniauxia*	n.m.	n.m.	n.m.	n.m.	n.m.	n.m.	n.m.	n.m.	0 *	0 *

## Data Availability

The data presented in this study are available within the article and Appendix A.

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
