# Peer review of "The Influence of the Grapevine Bacterial and Fungal Endophytes on Biomass Accumulation and Stilbene Production by the In Vitro Cultivated Cells of Vitis amurensis Rupr."

_plants, 2021, doi:10.3390/plants10071276_

Round 1

Reviewer 1 Report

Dear Author, 

the paper described the influence of bacteria and fungi on accumulation and on the production of stilbene. 

I've read the paper and I think that is well written. However, I have some consideration. 

In the introduction section, according to my point of view, a small paragraph should be added about stilbene and the importance of studying them (with some references). 

Always in the introduction, align the line spacing of the plant lines with the rest of the paragraph.

In the materials and methods section you write that the samplings were made in the months of June-September 2018-2020. It is not very clear to me how many samples have been taken and consequently if the endophytic bacteria and fungi have been isolated in different periods of plant development.

the section discussion, in my opinion, can be improved in the information.

Minor revision: 

correct the number of the section material and methods 

line 83: change font size 

line 251: specify that PDA is for fungi and R2A for bacteria as in the paragraph 4.3

line 271: add space between acid and constantly

line 275-376: it is not well written

Author Response

  1. “In the introduction section a small paragraph should be added about stilbene and the importance of studying them (with some references).” 

First, we would like to thank the Reviewer for the time and the constructive comments on our work. We added a paragraph describing the useful properties of stilbenes in the Introduction. See line 78.

  1. “Always in the introduction, align the line spacing of the plant lines with the rest of the paragraph.”

 Thank you for your comment. We corrected the line spacing in the Introduction.

  1. “In the materials and methods section you write that the samplings were made in the months of June-September 2018-2020. It is not very clear to me how many samples have been taken and consequently if the endophytic bacteria and fungi have been isolated in different periods of plant development.”

Thank you for your comment. The leaves and stems were collected from two grape plants located at a distance of 1 km from each other. The material was collected over 3 years 2018-2020, in June (young leaves and stems), July (mature leaves and stems) and in September (leaves that have become drier and changed color) 2018-2020. We made these clarifications in section 5.1. Plant material.

 “The section discussion can be improved in the information.”

Thank you for your comment. We have rewritten the Discussion chapter and added more information.

Minor revision:

  1. correct the number of the section material and methods
  2. line 83: change font size
  3. line 251: specify that PDA is for fungi and R2A for bacteria as in the paragraph 4.3
  4. line 271: add space between acid and constantly
  5. line 275-376: it is not well written

Thank you for your comments. We made corrections according to the list.

Reviewer 2 Report

The manuscript is interesting and provides new data The influence of the grapevine bacterial and fungal endophytes on the biomass accumulation and stilbene  production by the in vitro cultivated cells of Vitis amurensis Rupr. The subject of this manuscript is consistent with the scope of the Journal. There is no very important chapter: conclusions.  These results are interesting and very important for future of agricultural biotechnological. 
However, manuscript can be published in scientific Plants after some changes (major revision):

-    I wonder what the information on lines 74-81 is for, how are these results presented below?
-    Please correct the description of the material in accordance with the requirements of the journal and indicate how the biological material was transported? 
-    No information on the number of all experiments repetitions.
-    I think that the isolation efficiency should be confirmed by PCR with specific primers for bacteria and fungi.
-    Please let me know how many times it was done re-culturing of obtained strains. . 
-    I think that too short gene fragments were used: 16S rRNA and ITS, respectively, to identify bacteria and fungi. 
-    I am also wondering about the correctness of the DNA isolation method used, it was usually used to isolate DNA from plants.
-    The sequences according to the journal standards should be submitted to the GenBank database and these sequences will have a distinctive accession number.
-    The discussion chapter is not enough information.
-    Please, be sure that all the references cited in the manuscript are also included in the reference list and vice versa with matching spellings and dates.

Author Response

  1. “There is no very important chapter: conclusions. These results are interesting and very important for future of agricultural biotechnological“.

First, we would like to thank the Reviewer for the time and the constructive comments on our work. We added the Conclusions chapter to our manuscript.

  1. “I wonder what the information on lines 74-81 is for, how are these results presented below?”

Thank you for your recommendation. We partially deleted this information. We did not delete the rest of this paragraph to briefly explain the reader what the article is about.

  1. “Please correct the description of the material in accordance with the requirements of the journal and indicate how the biological material was transported? “

Thank you for your comment. We corrected the description of the material and methods. We added a description of how the material was transported to the laboratory in Chapter 5.1. Plant material. See line 296.

  1. “No information on the number of all experiments repetitions. “

Thank you for your comment. This information was provided in Chapter 5.7. Statistical Analysis. In the revised manuscript, we also added the information about the number of repetitions of experiments in Chapter 5.3. Treatment of grape cells with endophytic bacteria and fungi. See line 331. 

  1. “I think that the isolation efficiency should be confirmed by PCR with specific primers for bacteria and fungi. “

Thank you for your comment. We used primers which are wildely used for bacteria (16S) and fungi (ITS).

38.Lane, D.J. 16S/23S rRNA Sequencing. In: Stackebrandt, E. and Goodfellow, M., Eds., Nucleic Acid Techniques in Bacterial Systematic, John Wiley and Sons, New York. 1991, 115-175.

39.White, T.J.; Bruns, T.; Lee, S., Taylor, J. Amplification and direct sequencing of fungal ribosomal RNA genes for phylogenetics. PCR Protocols: A Guide to Methods and Applications. 1990, 315-322.

These references are given in the revised manuscript.

  1. “Please let me know how many times it was done re-culturing of obtained strains. “

Re-cultivation of the endophytic strains of bacteria and fungi was carried out once or twice before co-cultivation with the grape cells. We added this information to Section 5.3. Treatment of grape cells with endophytic bacteria and fungi. See line 330.

  1. “I think that too short gene fragments were used: 16S rRNA and ITS, respectively, to identify bacteria and fungi. “

We are very grateful for this helpful comment. In the revised manuscript, we used new primers to produce 1511 bp of 16S and we included the data in the Table (1). As to ITS, in the previous manuscript version, we mistakenly provided incorrect information about the primers used to amplify ITS and the size of the PCR products. We apologize for the incorrect information. Actually, primers were used which produce 592 bp of ITS. The corresponding changes were made in the chapter 5.2. Isolation and identification of endophytic bacterial and fungal strains. See line 312.

  1. “I am also wondering about the correctness of the DNA isolation method used, it was usually used to isolate DNA from plants.“

The method of DNA extraction used in this study has been widely proven not only for the isolation of plant DNA, but also for the isolation of bacterial DNA isolated from marine invertebrates.

  1. Kiselev, K.V.; Ageenko, N.V.; Kurilenko, V.V. Involvement of the cell-specific pigment genes pks and sult in the bacteria defense response of the sea urchin Strongylocentrotus intermedius. Dis. Aquat. Org. 2013, 103,121-132.

In any case, the quality of DNA isolation was checked by PCR. Using this method, we obtained and sequenced the PCR products of 16S and ITS.

  1. “The sequences according to the journal standards should be submitted to the GenBank database and these sequences will have a distinctive accession number.”

Unfortunately, getting sequence numbers in GenBank can take a long time. Therefore, we decided to provide files with the obtained nucleotide sequences of the studied strains to the supplementary material. Please see supplementary material for this manuscript.

  1. “The discussion chapter is not enough information.”

Thank you for your comment. In the revised manuscript, we have rewritten the discussion chapter in more detail.

  1. “Please, be sure that all the references cited in the manuscript are also included in the reference list and vice versa with matching spellings and dates.”

Thank you for your comment. We checked and carefully arranged the list of references.

Round 2

Reviewer 2 Report

Dear Authors,

Thanks for clarifying and improving the manuscript. Unfortunately, I disagree with this explanation.

  1. “The sequences according to the journal standards should be submitted to the GenBank database and these sequences will have a distinctive accession number.”

Unfortunately, getting sequence numbers in GenBank can take a long time. Therefore, we decided to provide files with the obtained nucleotide sequences of the studied strains to the supplementary material. Please see supplementary material for this manuscript.

This process does not take long and I believe it is required in order to be able to compare sequences. It is the specialists from the GenBank database that check whether the sequences are publishable. I believe this aspect is required.